# High Mobility Group AT-Hook 2 (HMGA2) Oncogenicity in Mesenchymal and Epithelial Neoplasia

**DOI:** 10.3390/ijms21093151

**Published:** 2020-04-29

**Authors:** Uchenna Unachukwu, Kiran Chada, Jeanine D’Armiento

**Affiliations:** 1Department of Anesthesiology, Columbia University Medical Center, 630 West 168th Street, P&S 12-402, New York, NY 10032, USA; uu2110@cumc.columbia.edu; 2Department of Biochemistry & Molecular Biology; Robert Wood Johnson Medical School, 675 Hoes Lane, Piscataway, NJ 08854, USA; chada@rwjms.rutgers.edu

**Keywords:** HMGA2 misexpression, chromosomal translocation, *let-7*, mesenchymal tumorigenesis

## Abstract

High mobility group AT-hook 2 (HMGA2) has been associated with increased cell proliferation and cell cycle dysregulation, leading to the ontogeny of varied tumor types and their metastatic potentials, a frequently used index of disease prognosis. In this review, we deepen our understanding of HMGA2 pathogenicity by exploring the mechanisms by which HMGA2 misexpression and ectopic expression induces mesenchymal and epithelial tumorigenesis respectively and distinguish the pathogenesis of benign from malignant mesenchymal tumors. Importantly, we highlight the regulatory role of *let-7* microRNA family of tumor suppressors in determining HMGA2 misexpression events leading to tumor pathogenesis and focused on possible mechanisms by which HMGA2 could propagate lymphangioleiomyomatosis (LAM), benign mesenchymal tumors of the lungs. Lastly, we discuss potential therapeutic strategies for epithelial and mesenchymal tumorigenesis based on targeting the HMGA2 signaling pathway.

## 1. Introduction

High mobility group AT-hook 2 (HMGA2) belongs to the HMGA family of small, non-histone chromatin-associated proteins [1]. This protein is encoded by the *HMGA2* gene localized to human chromosome 12 at band q14.3 [2], comprising five exons dispersed over a genomic region of ≥140 kb. Each of the first three exons contain conserved DNA binding domains called AT-hook motifs separated from an acidic C-terminal tail in the fifth exon by a spacer domain encoded by the fourth exon [1]. This structural feature determines HMGA2’s binding preference for AT-rich regions in the minor groove of DNA that causes ordered architectural changes which influence the conformation of bound DNA substrates, functional interactions between transcription factors, changes in chromatin structure, DNA replication, and gene transcription [3]. These physiological changes play fundamental roles in mammalian growth and development such that homozygous *Hmga2^−/−^* mice exhibit complete histological composition but yield a pygmy phenotype displaying dramatic reduction in adipose tissue accumulation and birth weight, and impairment of skeletal muscle development and myoblast proliferation [4,5]. Organ systems affected by *HMGA2* mutations thus highlight its potential role in fate specification of mammalian tissues to a mesenchymal lineage during embryonic development. This tissue lineage specificity for Hmga2 during development is further supported by evidence describing mesodermal differentiation, self-renewal and proliferation of human embryonic stem cells (hESCs) induced by HMGA2 expression [6].

Indeed, Hmga2 is ubiquitously expressed in undifferentiated tissues at early mammalian embryogenetic stages, beginning at 9.5 days post-coitum (dpc) in the mouse embryo and with time, expression becomes increasingly restricted to mostly undifferentiated tissue regions of mesenchymal origin, and some parts of the central nervous system [7]. Between developmental stages 14.5 – 17.5 dpc, the pattern of Hmga2 expression declines and is observed to be akin to the distribution of connective tissues in the mouse tissue mesenchyme, and further restricted to proliferative tissue regions [7]. As tissue differentiation progresses in the maturing human fetus, HMGA2 expression is also restricted to specific regions of the lungs, kidneys, and synovia [7,8], and associated with the activation and renewal of endogenous tissue-resident stem cells in adult stages [5,9]. These HMGA2-positive adult stem cells are possible undifferentiated tissue remnants of ontogenetic development, although more characterization studies of tissue-specific stem cell populations still need to be performed. Given that HMGA2 expression dynamics are functionally superimposable to mammalian embryogenic differentiation paradigms, genetic anomalies at the HMGA2 locus during this tissue maturation period could account for anomalous cell fate specification, which could lead to multi-systemic neoplasms, determining tissues that would become tumorigenic, and the timing of tumor ontogeny.

The HMGA family of proteins was first isolated from cancerous HeLa S3 cells in 1983 [10]. However, the correlation between HMGA2 and neoplastic transformation was not established until two years later, when HMGA nuclear phosphoproteins were detected in a rat thyroid cell line (FRTL5) after viral transformation [11]. The isolated HMGA proteins were associated with a highly malignant phenotype irrespective of whether transformed cells were chemically, virally or spontaneously derived [12]. More direct evidence for the oncogenic role of HMGA proteins was reported when rat and human cell lines with ectopic expression of *HMGA1* transcript variant (*HMGA1a*) and *HMGA2* formed tumors and led to distant metastases when injected in athymic nude mice [13]. Since then, numerous postulates have been put forward and experiments conducted to explain the causative biological mechanisms employed by HMGA proteins to induce both benign and malignant neoplasms. These mechanisms of neoplastic transformation have been found to be tumor-type specific and to differ between epithelial and mesenchymal tumors. These mechanisms are discussed in subsequent sections of this review. The lack of HMGA2 expression in proliferating fibroblasts of some pulmonary interstitial diseases further suggests that the gene’s misexpression defines neoplastic transformation of a normal cell rather than a hyper-proliferative index [14].

Summarily, ectopic HMGA2 expression drives epithelial tumor metastasis and multiplicity in cell culture and in in vivo mouse models mainly by the activation of the TGFβ pathway and epithelial–mesenchymal transitions (EMT) [15,16]. In contrast, the misexpression of full length, chimeric or truncated *HMGA2* mRNA transcripts in differentiated benign mesenchymal tumors derived from abnormal chromosomal breaks governs mesenchymal tumorigenesis irrespective of the nature of the *HMGA2* gene product [17,18]. In this review, we will explore these mechanisms by which HMGA2 induces epithelial and mesenchymal tumorigenesis, and discuss the gene’s role in lymphangioleiomyomatosis, a rare pulmonary mesenchymal neoplasm of unknown etiology that is often a clinical manifestation of tuberous sclerosis. Lastly, we identify potential therapeutic strategies for epithelial and mesenchymal tumorigenesis based on targeting the HMGA2 signaling pathway.

## 2. Mechanisms of HMGA2-Induced Mesenchymal Tumorigenicity

From our review of cytogenetic studies (see Appendix A), causative mechanisms of *HMGA2*-induced mesenchymal tumorigenesis results from HMGA2 misexpression in well differentiated mesenchymal tissues contrary to its canonical expression in undifferentiated mesenchyme [7,8]. These genetic changes enhance or repress genes and transcription factors that play crucial roles in cell proliferation [19], cell cycle regulation [20,21], DNA damage response [22,23,24], apoptosis [25] and cellular senescence [26,27], all causatively leading to tumor phenotypes in mesenchymal tissues [28]. In this section, we explore evidence for HMGA2-induced pathogenesis in benign and malignant mesenchymal neoplasms.

### 2.1. HMGA2 Misexpression

HMGA2 is canonically expressed in the undifferentiated mesenchyme and is undetectable in the differentiated tissue forms [8]. However, in several types of differentiated mesenchymal tumor, including lipomas [29,30], leiomyomata [8,31,32,33], pulmonary chondroid hamartomas [34,35], endometrial polyps [36], and soft tissue chondromas [37], certain genetic mechanisms enable its untimely re-expression, biologically termed HMGA2 misexpression. A variety of *HMGA2* transcripts have been isolated from these tumors and believed to cause mesenchymal tumorigenesis through a re-expression of the *HMGA2* gene in differentiated tissues. Some of these transcripts derive from chromosomal translocations at 12q13–15 which truncates the human *HMGA2* open reading frame (ORF) resulting in loss of its C-terminus and/or fusion to partner ectopic sequences [38,39], while in other mesenchymal tumors, intact ORF have been resolved but with truncated 3’ untranslated region (UTR) which encodes binding sites for *let-7*, a negative regulator of HMGA2 expression [40,41]. These chromosomal translocations have been reported as the genetic mechanisms necessitating HMGA2 misexpression in mesenchymal tumors [17,42,43]. Similar translocations involving the chromosomal locus 6p21–23 corresponding to HMGA1 (HMGI-Y) have also been described in benign mesenchymal tumors, implicating the HMGI family of DNA-binding proteins [44,45].

Chromosomal abnormalities are a hallmark of cancers, and their causes, although poorly understood, are commonly attributed to environmental and occupational exposures and certain therapies [46,47]. Chromosomal abnormalities can occur in the form of balanced chromosomal rearrangements such as reciprocal translocations or inversions that result in the formation of chimeric fusion genes, predicted to be early initiating events in tumorigenesis [47]. With non-balanced chromosomal rearrangements, some genetic material is lost or gained by deletions or duplications resulting in reduction or enhancement of genetic activity respectively [47]. We reviewed cytogenetic analyses of mesenchymal tumors and found that most chromosomal abnormalities occurred by balanced chromosomal rearrangements involving *HMGA2* locus 12q13–15 [48,49]. A listing of these chromosomal rearrangements involving *HMGA2* in human mesenchymal tumors is catalogued in Appendix A. Our data compilation reveals that the most common translocation partners of *HMGA2* in mesenchymal tumors are chromosomes 1–3, with a preponderance of intragenic chromosome 12 aberrations. In addition, certain tumor types were synonymous with *HMGA2* gene translocations involving specific chromosomes, as observed with chromosome 14 in uterine leiomyomata and pulmonary chondroid harmatomas (Appendix A).

The nature of the chromosomal rearrangements and resulting mRNA *HMGA2* transcript also differ by mesenchymal tumor type [50]. In most lipomas and pulmonary chondroid harmatomas, chromosomal translocations have breakpoints that preferentially cluster in the third intron of the *HMGA2* gene to yield either oncogenic truncated forms of *HMGA2* mRNA transcripts containing exons 1–3 and lacking the 3’ untranslated region (3’ UTR), and/or chimeric fusion transcripts co-joining truncated HMGA2 DNA binding domains to up- or downstream transcriptional regulatory sequences [51,52]. Common HMGA2 fusion partners are tumor suppressors whose functions are typically lost following these gene rearrangements [53,54]. On the other hand, in most uterine leiomyomata, these chromosomal breakpoints mostly occur 10–100 kilobases upstream of the HMGA2 coding region such that full-length gene transcripts are expressed in these tumor types with or without chimeric forms [41,50,55,56].

Overall, these chromosomal translocations lead to the misexpression of three main types of *HMGA2* transcript: 1) full-length *HMGA2* transcripts with no apparent disruption of the coding sequence; 2) truncated *HMGA2* transcripts lacking the 3’-UTR; and 3) chimeric *HMGA2* transcripts fused to other ectopic sequences. Some mesenchymal tumor types were found to express only truncated, chimeric or full-length *HMGA2* transcripts, while in other mesenchymal tumors, multiple *HMGA2* transcript types could be found, indicating that the nature of *HMGA2* disruption in these tumors did not affect the gene’s transformative capability [17].

#### 2.1.1. Full Length *HMGA2* Transcript with No Apparent Disruption of the Coding Sequence

Despite previous studies showing that truncation and/or addition of ectopic sequences after the third AT-hook were necessary for neoplastic transformation of murine fibroblasts [57], many human uterine leiomyomata, mammary fibroadenomas, and salivary gland adenomas were found to harbor full-length *HMGA2* coding regions [8,40,41,58]. Full length *HMGA2* transcripts can form in these tumor types when chromosomal rearrangements occur in an extragenic location usually upstream of the HMGA2 coding region [17,43,59]. In human uterine fibroids, such chromosomal rearrangements often involved the *HMGA2* gene and chromosome 14, and balanced intragenic chromosome 12 aberrations were rare [41]. These findings suggest that full-length HMGA2 has oncogenic properties, and that disruptions in regulatory elements proximal to the *HMGA2* coding region could lead to *HMGA2* misexpression, inducing mesenchymal tumorigenesis [41]. Consistently, misexpression of full-length Hmga2 induces benign mesenchymal tumors in mice [17].

Another mesenchymal tumorigenic mechanism allowing for the expression of full length *HMGA2* transcripts has been described in well differentiated liposarcomas (WDLPS) and atypical lipomas (ALP) [29,60]. In these mesenchymal tumors, supernumerary rings and giant rod marker chromosomes comprised various amplified subregions of different chromosomes associated with 12q14–15 [29,60]. In addition to ectopic expression of *HMGA2*, other genes co-amplified as a result of these supernumerary structures include *MDM2*, *CDK4* and *TSPAN31*. However, supernumerary ring chromosomes are a rare find in ordinary lipomas [60].

#### 2.1.2. Truncated *HMGA2* Transcript Lacking the 3’-UTR

The HMGA2 3’ UTR has considerable potential for posttranscriptional regulation by RNA-binding proteins and miRNA-induced silencing complexes (miRISCs) [61,62]. Chromosomal rearrangements and intragenic chromosomal breakpoints within the region 12q14~15 leading to the formation of truncated *HMGA2* transcripts lacking the 3’ untranslated region (UTR) of *HMGA2* mRNA have been observed in many benign mesenchymal tumors, including some uterine leiomyomata and pulmonary chondroid hamartomas [62,63]. Human *HMGA2* 3’ UTR is 2.9kb long and harbors up to 35 discrete positive and negative cis-regulatory elements that act independently, or less commonly in synergy, altogether functioning to repress *HMGA2* expression post-transcriptionally [64]. Although most regulatory elements in the *HMGA2 3’ UTR* have been found to induce HMGA2 expression such as AU-rich elements (AREs) that interact with HuR ARE-binding protein [65], repressive regulatory elements include characteristic binding sites for the tumor suppressor microRNA *let-7* family [62,64,66]. The role of *let-7* in HMGA2-induced mesenchymal tumor pathogenesis is further discussed below.

#### 2.1.3. Chimeric *HMGA2* Transcript Fused to Ectopic Sequences

Translocation breakpoints in the chromosome 12 locus of the *HMGA2* gene preferentially occurs in the large intronic space linking exons 3 and 4 resulting in a truncated gene with intact DNA and protein binding domains but lacking a carboxy terminus including the 3’ untranslated region (3’ UTR) [39,67]. These aberrant *HMGA2* transcripts can fuse to ectopic sequences from translocation partners whose protein functions are frequently lost due to gene rearrangements [48] (Figure 1E). Table 1 lists the balanced chromosomal rearrangements in mesenchymal tumors that form chimeric *HMGA2* fusion transcripts. Supporting previous cytogenetic reports, Table 1 shows that the lipoma preferred partner gene (LPP) was one of the most abundant fusion gene partners to *HMGA2* in lipomas, in a case of pulmonary chondroid harmatomas, and in soft tissue chondromas, and involved translocation loci t(3;12)(q14~15;q12~21) [29,37,39,68,69,70,71,72,73,74]. The *HMGA2-LPP* gene codes for a transcription factor containing the AT-hook domains of HMGA2 fused to three LIM domains at the C-terminal [75]. The expression of the HMGA2-LPP fusion protein typically leads to neoplastic transformation although the expression of the fusion gene did not increase the transformative ability of the truncated HMGA2 [57,75]. Interestingly, another fusion gene, the SET binding protein 1 (SETBP1), encoding a protein that binds the SET nuclear oncogene involved in DNA replication, was also found to recur in lipomas and osteochondrolipomas (t(12;18)(q27~28;q13~15)) [42]. *SETBP1* was fused to truncated *HMGA2* containing exons 1–3, or to an intragenic sequence 18q12.3 that is 10kbp distal to *SETBP1* [42]. In the *HMGA2-SETBP1* chimeric transcripts, the translocation breakpoint in *SETBP1* occurred at the 3’-UTR essentially deregulating *HMGA2* and possibly influencing expression of SETBP1 [42]. Of note, intragenic chromosomal breakpoints predominated in sarcomas where in almost all cases investigated, chimeric fusion *HMGA2* transcripts were formed [76]. On the contrary, in lipomas where rearrangements of the *HMGA2* gene involve multiple chromosomal partners, chimeric fusion genes are rarely observed (Appendix A).

#### 2.1.4. *let-7* Regulation of HMGA2 Misexpression in Benign Mesenchymal Tumors

As all forms of *HMGA2* transcript exhibit similar transformative potential, it can be hypothesized that a single regulatory regimen could chiefly govern the misexpression of these different transcript types and their propensity to drive the pathogenesis of benign mesenchymal tumors [18]. One such regulatory element is the *let-7* microRNA family of tumor suppressors which have conserved complementary binding sites encoded in the 3’-UTR of the *HMGA2* gene [61,62]. *Let-7* expression is inversely related to HMGA2 expression, is undetectable during embryogenesis (Figure 1A), but increases after differentiation and in mature tissues [62]. In line with established models of miRNA action in mammalian cells [92], in the normal differentiated mesenchyme, *let-7* post-transcriptionally recognizes its target sites in the full-length *HMGA2* mRNA and recruits it to exonucleases leading to target mRNA decapping and degradation of the HMGA2 gene transcript (Figure 1B). This would account for the reciprocal relationship between *HMGA2* and *let-7* expression [62]. These *let-7*-*HMGA2* molecular associations can further specify tissue-type differentiation of normal mesenchymal tissues, selectively enhancing human osteogenesis while repressing adipogenesis [93].

In uterine leiomyomas, the inverse physiological relationship between *let-7* and HMGA2 also persists such that large leiomyomata were observed to express low levels of *let-7* and high levels of HMGA2, while small leiomyomata expressed high levels of *let-7* and low levels of HMGA2 [94]. These findings establish a direct correlation between endogenous HMGA2 and *let-7* levels and suggest that additive binding of *let-7* to 3’ UTR complementary sites in *HMGA2* could progressively decrease protein translation and HMGA2 pathway activation, limiting tumor size. Given the multiple binding sites for *let-7* in many chromosomal 3’ UTR loci, it is possible that HMGA2 is mis-expressed when the relative abundance of *HMGA2* mRNA transcripts suffice to bind available *let-7* and sustain HMGA2 re-expression in benign mesenchymal tumors. High levels of *HMGA2* transcription in larger leiomyomata could encode more 3’ UTR binding sites to soak up available mature *let-7* miRNA limiting the repressive and degradative function of *let-7* on HMGA2 expression [40,41], and allowing for greater rates of tumorigenesis compared to smaller sized uterine fibroids. This potential tumorigenic mechanism is depicted in Figure 1C and could explain the expression of full length *HMGA2* mRNA transcripts in these uterine leiomyomata. This mechanism could also account for slight differences in the tumor spectrum mediated by full length forms of *Hmga2* versus truncated *Hmga2* forms lacking the *let-7* binding sites in mice [18,95]. Ectopic expression of truncated *HMGA2* transgenes in an immortalized mesenchymal stem-like cell line stymied adipogenic differentiation and upregulated genes for transcription and intracellular protein transfer compared to wildtype *HMGA2* indicative of higher tumorigenicity and lesser *let-7* repressive activity in cells expressing truncated HMGA2 [96].

In many other mesenchymal tumors, activated LIN28A or LIN28B RNA-binding protein homologues directly interact with the terminal loop region of either *pre-let-7* and/or primary *let-7*, preventing their biogenesis and tumor repressive function, and can induce their degradation [97,98,99]. Additionally, by repressing *let-7*, LIN28A/B indirectly upregulates cell cycle regulators targeted by *let-7* such as cyclinD1/2, CDK6, CDC34, CDC25A, and TRIM71, and cell proliferation pathway targets PI3K/AKT, MAPK, MYC, RAS and BLIMP1 leading to aberrant proliferation of tumor cells [99]. Conversely, *let-7* can also bind to complementary sites in the 3’ UTR of LIN28A/B mRNAs inhibiting their expression and function [100]. Other RNA-binding proteins like IMP3 exist in stable cytoplasmic granules which physically associate with and protect *HMGA2* mRNA from *let-7*-dependent degradation [101]. 

Most frequently, chromosomal break points truncate the *HMGA2* 3’ UTR in mesenchymal tumors and can thus prevent the docking of *let-7* to its binding sites and the canonical repression of HMGA2 expression in differentiated mesenchymal tumors [62] (Figure 1D). Truncated HMGA2 also occur as chimeras fused to ectopic sequences by chromosomal translocations (Figure 1E). It has been suggested that the loss of *let-7* complementary sites in *3’UTR* stabilizes *HMGA2* mRNA in some of these tumors [63] leading to a deficiency in *let-7*-mediated regulation [61], increased cell proliferation and tumorigenesis [43,102]. These studies highlight the central role *let-7* expression plays in multiple mechanisms of HMGA2 misexpression causing mesenchymal tumorigenesis. Given that multiple genes distributed throughout the human genome encode for *let-7*, loss of the 3’UTR might not suffice to explain all HMGA2-mediated tumorigenic events in the mesenchyme [62]. Indeed, in a recent study, *let-7* accounted for only 15% of the total regulatory effects determining HMGA2 expression/misexpression [64]. This seemingly limited influence of *let-7* in directly regulating HMGA2 misexpression could be more reflective of its cooperative roles with other translation regulatory networks in the ontogenesis of mesenchymal tumors. In mechanistic studies using adipogenic progenitor cells of lipomas and uterine leiomyomas, a p14^Arf^-MDM2-TP53-let-7 network stabilized HMGA2 expression, lowering the tendency for malignant transformation while maintaining stem-like proliferative characteristic of benign mesenchymal tumor cells [26]. p14^Arf^ repressed HMGA2 via a TP53 mechanism that has been linked to increased expression of *let-7*, whereas FGF1-stimulated increase in HMGA2 increased p14^Arf^ [26,103]. These study results are indicative of a bimodal mechanism of regulating cellular senescence and stem-like cell renewal forming the pathobiological basis for benign mesenchymal tumorigenesis [9].

Unlike the preponderance of studies performed on benign mesenchymal tumors, only a few preliminary studies have determined the role of HMGA2 misexpression in malignant mesenchymal tumors such as well differentiated liposarcomas and osteosarcomas [37,60,76,91]. This may suggest that HMGA2 misexpression is a relatively rare event in malignant mesenchymal tumors. In addition to canonical HMGA2 misexpression presently described, these tumors commonly possess supernumerary ring structures and giant rod chromosomes exhibiting gene amplifications at the 12q13–15 *HMGA2* loci and in proximally located genes such as *MDM2* [59,104,105].

### 2.2. Effects of HMGA2 Misexpression on Chromatin Structure

Accessibility to DNA within chromatin remains central to the epigenetic regulation of eukaryotic DNA-dependent nuclear processes such as transcription, replication, recombination, and repair [106]. Nucleosomes are the structural and functional units of chromatin, comprising DNA surrounding histone octamers to form nucleosome cores that are joined by linker DNA [107]. Two major epigenetic regulatory processes—post-translational modification of histones and chromosome remodeling—yield chromosomal conformations that allow transcriptional access to DNA in chromatin [108]. Some of the post-translational modifications necessitating architectural changes in chromatin include DNA methylation and histone acetylation [108,109,110]. As an example, in gliomas, HMGA2 was shown to form a complex with histone lysine acetyltransferase GCN5 and bind to AT-rich promoter region of matrix metalloproteinase 2 (MMP2), catalyzing the histone acetylation and chromatin conformational remodeling of the promoter that induced gene transcription and invasive phenotype of glioblastoma cells [111].

In addition, H1 histones can bind to linker DNA and increase the compactness of chromatin, providing a barrier to sequence-specific recognition sites on DNA [112,113]. In turn, the HMGA2 and HMG families of proteins also bind to nucleosomes and functional AT-rich motifs in DNA minor groove, competing with H1 histones for binding to linker DNA [114,115]. In doing so, HMGA2 modulates H1 histone binding to chromatin, replacing post-translationally modified H1 histones in some cases [106,116]. This induces loosening and conformational changes in the chromatin structure, that affects nucleosome accessibility in both genetically active euchromatin and inactive heterochromatin [106,116]. This ability of HMGA2 to constrain chromatin supercoiling has been shown to confer protection and genome stability in human fibrosarcoma, embryonic stem cells (HESCs), lung epithelial cells, and adenocarcinomas during DNA replication [22,117,118]. HMGA2 inhibits replication fork regression and cytotoxic double stranded breaks (DSBs) generated by chemotherapeutic agents (anti-topoisomerases) and/or during excessive DNA supercoiling [22,117,118]. As such, at high levels of expression, HMGA2 is a critical determinant of tumor response to chemotherapy and tumor cell survival and invasiveness [117,119].

For complete access to transcriptional regulatory elements on target genes, chromatin remodelers such as the FACT complex [120] and the anti-silencing function 1 histone chaperone (ASF1) [121] will be necessary participants. These protein complexes have been shown to facilitate eviction/deposition of histones from nucleosome cores, destabilizing/stabilizing nucleosome structure for transcription elongation processes [120,121]. Given that HMGA2 also binds to these chromatin remodeling protein complexes, a role of HMGA proteins in the dynamics of nucleosome core architectures during transcriptional regulation has been suggested [106]. These HMGA2–nucleosome associations help establish cell identity, and initiate tissue differentiation programs, such as the EMT-driven epithelial tumorigenesis [106] discussed in the subsequent section. However, causative mechanisms of mesenchymal tumorigenesis by HMGA2-induced alteration of chromatin structure is yet to be delineated for most mesenchymal tumors.

### 2.3. HMGA2-Induced Tumorigenesis in Lymphangioleiomyomatosis

Lymphangioleiomyomatosis (LAM) is the major pulmonary manifestation of the tuberous sclerosis complex disorder (TS) presenting as benign mesenchymal tumors and lesions that cause recurrent pneumothorax, lung cysts, chylous pleural effusions and renal and abdominal angiomyolipomas (AMLs) [122,123,124,125,126]. This disorder occurs either sporadically (S-LAM), or as an autosomal dominant inheritance of mutations in the *Tsc1* or, more frequently, *Tsc2* tumor suppressor genes (TSC-LAM) [126,127,128,129,130,131,132]. Heterozygous *TSC1/2* mutations have been causatively linked to hyperactivation of the mammalian target of rapamycin (mTOR) pathway, leading to the aberrant cell growth and proliferation characterizing benign tumors in LAM [133,134,135,136,137,138]. However, as many LAM patient tumors do not exhibit TSC mutations especially in S-LAM cases [139], biochemical signaling due to activation of the mTOR pathway does not solely account for tumorigenesis in LAM [138,140,141]. This explains why pharmacological inhibition of mTOR pathway targets is not curative [142,143,144], and in most treatment cohorts, disease symptoms tend to recur upon discontinuation of therapy [145]. It is thus imperative that alternative etiologies to LAM pathogenesis in addition to genetic mechanisms at the *TSC1/2* loci be considered.

In our laboratory, we focused on the mechanisms of HMGA2 in LAM pathogenesis, and established that although HMGA2 was mis-expressed in 100% of tumors resected from LAM patients in our studies and from *Tsc2^+/−^* mice, an established animal model for LAM [14,146], *Hmga2^−/−^Tsc2^+/−^* mice exhibited minimal renal (epithelial) tumors and no extra-renal (mesenchymal) tumors) [146]. These results indicate that Hmga2 expression is absolutely required for mesenchymal tumorigenesis in the *Tsc2^+/−^* mice. We further observed analogous expression of IGF2BP2, an oncofetal protein and downstream target of HMGA2 in 100% of human LAM and TSC lesions [146], similar to the gene’s pattern of co-expression with HMGA2 in rhabdomyosarcoma and during myoblast proliferation [5,147]. Importantly, the tuberin protein product of *Tsc2* expression was present in all *Tsc2^+/−^* mouse and human mesenchymal tumors, and about 80% of *Tsc2^+/−^* mouse renal carcinoma [146]. Adding that only 50% of human mesenchymal tumors, and 31% of *Tsc2^+/−^* mouse renal tumors exhibit altered mTOR pathway activation, supports the existence of tumorigenic mTOR-independent mechanisms causing LAM pathogenesis [146]. The lack of HMGA2 expression in similar proliferative interstitial lung diseases to LAM, like interstitial pneumonitis and pulmonary fibrosis, also suggests that HMGA2 misexpression transforms tumor cells in LAM, and is not solely due to an abnormal increase in cell proliferation rates [14].

However, there has been a dearth in studies investigating the mechanistic potential of HMGA2-induced neoplasms in LAM. As LAM pathogenesis is characterized by benign tumors with differentiating characteristics towards a mesenchymal phenotype, we postulated that the oncogenic triangle involving LIN28-*let-7*-HMGA2 observed in most mesenchymal tumors might account for the tumorigenic pathway employed by HMGA2 in LAM. Using an established *Tsc2^+/-^* mouse model of LAM [148], we have assessed the expression of *Hmga2* pathway targets in both mesenchymal and renal (epithelial) tumors. Compared to matched normal tissues, our results reveal significant downregulation of *let-7a* expression and upregulation of *Lin28a*, *Lin28b* and *Igf2bp2* in all mesenchymal tumors assessed including hepatic hemangiosarcomas, foot lymphomas, and hemangiomas, and in pulmonary adenomas by RT-PCR (unpublished data). In these mice, the absolute necessity for the repression of *let-7a* was observed for both epithelial and mesenchymal tumors supporting previous findings of the key role of the *let-7–Hmga2* axis in these carcinogenic processes [63,99]. Cytogenetic studies in the *Tsc2^+/−^* mouse mesenchymal tumors will determine whether chromosomal rearrangements play a pathogenic role in these LAM mouse models.

## 3. Mechanisms of HMGA2-Induced Epithelial Tumorigenicity

Nascent epithelial tumors and early adenomas typically exhibit undetectable or low levels of *HMGA2* mRNA [149] and low to moderate levels of the HMGA2 protein product [15]. However, an increasing expression of HMGA2 was observed to be distributed from non-neoplastic, well-differentiated centers of human colorectal tumors, human squamous carcinoma, and MMTV-Wnt1 transgenic mice mammary tumors towards the extremity invasive front where the overwhelming majority of tumor cells were HMGA2-positive [15,150]. In mammary and colorectal tumors, these invasive HMGA2-positive tumor cells exhibited membrane-to-nucleus re-localization of β-catenin, loss of E-cadherin, increase in levels of vimentin, and an analogous expression of HMGA2 downstream target IGF2BP2 [15,151]. This observed cytological transition in phenotype known as epithelial-mesenchymal transition (EMT) is a highly conserved gradual process that also occurs during embryonic development, histogenesis, and wound repair, and is precursory to tumor invasion and malignancy [15,152,153]. Indeed, HMGA2 expression increased proportionally with anchorage-independent growth and metastasis of many epithelial tumors including colon, gastric, and breast cancer cells [15,154,155]. In mammary tumor cell lines stably expressing HMGA2 by vector transfection, primary tumors formed at a faster rate and exhibited higher metastatic potential to the liver parenchyma and lungs than tumor cells with empty vectors after inoculation in mice [15]. In many studies, EMT has been reported to be the key mechanism for HMGA2-induced tumorigenesis in epithelial tissues [16,156,157].

How does ectopic expression of HMGA2 induce EMT for epithelial tumorigenesis, metastasis and invasion? In both in vitro and in vivo models of epithelial tumorigenesis, HMGA2 has been observed to activate the TGFβ pathway via TGFβRII leading to the phosphorylation and translocation of Smad3 from the cytoplasm to the nucleus [15], a mechanism necessary for EMT initiation [157,158]. For HMGA2 to induce epithelial tumorigenesis and invasiveness via EMT, it is necessary that TGFβ pathway is activated [15,157]. It is suggested that epithelial tumor cells, upon HMGA2 induction, might become more responsive to the TGFβ ligand typically present in tumor microenvironments, signaling for increased tumor metastasis [15]. Indeed, downstream of SMAD proteins, HMGA2 was resolved to directly activate zinc-finger transcription families Snail, Slug, and Twist and downregulate Inhibitor of differentiation 2 (Id2) in mammary tumor cells which altogether are known to repress E-cadherin (CDH1) expression [157,159,160]. In addition, ectopic HMGA2 expression also remodels chromatin in mammary tumor and breast cancer to a closed conformation at the Cdh1 locus by hypermethylation governed by HMGA2 interaction with DNA methyltransferases (DNMT3A) [161]. This biochemical mechanism represents a second layer of epigenetic control of E-cadherin expression during EMT-induced epithelial tumorigenesis.

Other small molecule regulators including microRNA and RNA-binding proteins also govern epithelial tumorigenesis induced by HMGA2. The *let-7* miRNA family have been shown to exert repressive control over HMGA2-induced epithelial tumorigenesis by binding to the 3’-UTR of human *HMGA2* gene [162]. In the course of EMT advancing carcinogenesis, ovarian cancer cells expressing an epithelial gene signature exhibited significantly higher levels of seven of the twelve members of the *let-7* family compared to cells defined by mesenchymal genes [162]. Individual *let-7* family members have also been implicated in a variety of tumor pathogenic mechanisms involving HMGA2. For instance, *let-7c* was shown to suppress EMT and proliferation of head and neck squamous cell carcinoma by targeting HMGA2 and IGF1R [163], while *let-7a* knockdown exhibited an analogous inhibitory function in nasopharyngeal carcinomas induced by HMGA2 expression, decreasing HMGA2 and the expression of EMT marker genes Snail, Slug, and vimentin [164]. However, LIN28 RNA-binding protein, serves as a competitive inhibitor of *let-7* binding preventing its maturation by inducing terminal uridylation and degradation of *let-7* precursors, and can thus de-repress HMGA2 expression, restoring tumorigenicity and invasiveness of epithelial tumors [165]. The LIN28-*let-7*-HMGA2 signaling axis is further modulated by Raf-1-kinase inhibitory protein (RKIP), a metastasis suppressor which induces expression of epithelial miR-200b that directly inhibits Lysyl Oxidase (LOX) expression leading to decreased transcription of *LIN28*, elevated *let-7* expression and inhibition of HMGA2 [166,167]. In another study of prostate cancer metastasis, overexpression of BTB and CNC Homology 1 (*BACH1*) transcription factor led to a significant decrease in *let-7A* expression and subsequent increase in HMGA2 which facilitated metastasis by promoting EMT [168]. Given the genomic ubiquity of *let-7* in regulating HMGA2 expression [62], further studies are required to clearly distinguish the role of *let-7* in mesenchymal versus epithelial tumorigenesis.

Ectopic expression of HMGA2 can also promote epithelial tumor cell proliferation and metastasis by influencing the cell cycle in a tumor cell-type dependent manner, where, for instance, its knockdown arrests ovarian cancer cells at G1 [169] and G2/M arrest for leukemia cells [19]. In some instances, HMGA2 can exert tumorigenic effects on cell cycle by directly inducing cyclin A2 [20], activating protein-1 (AP1) expression [170], and MDM2-mediated p53 ubiquitination [171] facilitating cell proliferation. Indirectly, HMGA2 can activate phosphatidylinositide 3-kinase (PI3K)/AKT/mTOR/p70S6k signaling which inhibits tumor suppressors p16^INK4A^, p21^CIP1/WAF1^ [172], retinoblastoma protein (pRB) [173] and p14^Arf^ [9] to enable transitions through cell cycle checkpoints, facilitate cell proliferation and restrain cellular senescence. During these replicative cycles, HMGA2 has also been postulated to either inhibit DNA damage response (DDR) mechanisms that ensure genomic stability at replication forks, which leads to increased DNA mutational rates at onset of tumorigenesis, or augment these DDR mechanisms to reduce replication recovery times after replication fork arrest in stem and cancer cells [22,174]. HMGA2 can also mediate epithelial tumorigenesis by modulating apoptosis in cancer cells. Breast adenocarcinoma cells overexpressing HMGA2 exhibit fewer apoptotic events compared to cells with low HMGA2 expression by a mechanism involving HMGA2-mediated inhibition of miR-34a and subsequent de-repression of Bcl-2 [25]. HMGA2 also protects cancer cells from apoptosis by hyperactivating the PI3K/Akt pathway, which impairs the activation of caspase-9 and Bad in a gastric cancer cell line [175].

Dynamic intracellular localization processes of *HMGA2* mRNA transcripts and translation products have been reported to also influence epithelial tumorigenesis and metastatic potential. Hmga2 expression has been detected in the cell membrane of quiescent non-transformed, post-natal mouse keratinocytes but upon onset of proliferation, membrane-to-nuclear translocation of the protein was observed [176]. Analogous Hmga2 nuclear translocation was also been observed upon ex vivo culture of mouse keratinocytes and during cutaneous carcinogenesis in DMBA and TPA mouse models where Hmga2 induces its own expression in an autoregulatory loop by binding to the *Hmga2* promoter [176]. Normal human epithelial prostate cell also exhibited low levels of HMGA2 expression in the plasma membrane which switched to predominantly cytoplasmic then nuclear localization with increasing prostate tumor grade, metastatic potential, and HMGA2 expression [177]. Specific small non-coding circular RNAs (circNSUN2) promotes EMT and human and mouse colorectal cancer (CRC) cell metastasis by stabilizing high levels of cytoplasmic *HMGA2* RNA-protein interactions in a circNSUN2/IGF2BP2/HMGA2/ complex and activating the HMGA2 pathway [178]. In this study, lower levels of cytoplasmic HMGA2 mRNA were associated with lower liver metastasis in CRC [178]. HMGA2 mRNA cytoplasmic stability is similarly governed by higher order ribonucleoprotein complexes formed with RNA binding proteins as observed in solid cancers where for instance, *HMGA2* mRNA is physically associated with IMP3 RNP in vivo [179]. There is a need for studies of these HMGA2 intracellular translocation processes and their relevance in mesenchymal tumorigenesis.

These studies make it clear that HMGA2 oncogenic mechanisms differ between mesenchymal tumors and their epithelial counterparts and even between tissue-specific tumor subtypes. HMGA2-induced mesenchymal tumorigenesis recapitulates the gene’s embryonic expression signature in mature differentiated tissues using various mechanisms leading to neoplastic growth [17,18,44]. In epithelial tumors, HMGA2 drives epithelial-to-mesenchymal cell transformations (EMT) towards a tumorigenic phenotype [15].

## 4. Therapeutic Considerations for HMGA2-Induced Neoplasia

High expression of HMGA2 has been associated with highly malignant phenotypes described by resistance to chemotherapeutic agents and metastases [180,181]. Along with *let-7* expression, HMGA2 has been determined to be a predictive biomarker of poor clinical outcomes in many epithelial and hematopoietic malignancies including acute myeloid leukemia [182], ovarian and colorectal carcinoma [162,169], and oral squamous cell carcinoma [183]. In some neoplasia, however, disease severity does not always correlate with increased HMGA2 expression levels [184,185], necessitating the resolution of other biomarkers in the HMGA2 pathway. Downstream of HMGA2, IGF2BP2 was found to be overexpressed and correlative to poor survival and induction of EMT in pancreatic ductal adenocarcinoma [186]. Given that many interacting partners in the HMGA2 pathway, including up- and downstream genes IGF2BP2, LIN28 and *let-7* have been elucidated, they serve as attractive targets whose modulation could be curative for many cancers involving the HMGA2 pathway. However, most therapeutic strategies adopted to probe targets in this pathway have focused chiefly on perturbing HMGA2 expression in epithelial neoplasms with limited success.

Gene silencing therapy using siRNA suppressed proliferation and growth of ovarian cancer cell lines overexpressing *HMGA2* by cell cycle arrest at G1 phase, and decreased the size of tumor xenografts in athymic nude mice treated with a *Hmga2*-targeting construct [169]. A similar effect was observed when siRNA- and miRNA-mediated silencing of *HMGA2* induced apoptosis, G2/M cell cycle arrest, and suppressed proliferation and invasion of human colorectal carcinoma [187,188]. Additionally, p53-induced *miR-1249* expression was antagonistic to HMGA2 expression and inhibited HMGA2-induced invasiveness of colorectal cancer cells by stabilizing the epithelial phenotype, decreasing expression of N-cadherin and vimentin and increasing E-cadherin expression [189]. Sustained expression of the epithelial phenotype was also observed in *HMGA2^-/-^* prostate cancer cells in which EMT was also inhibited [190]. Similarly, perturbation of HMGA2-HOXA9 signaling arrested the differentiation of human myeloid leukemia cells towards a pathogenic phenotype [191].

In another therapeutic strategy, small molecule inhibitors such as netropsin were found to block the binding of HMGA2 to DNA minor groove in AT sequences in a biosensor-surface plasmon resonance assay designed to screen for potent HMGA2 inhibitors [192]. However, a recent study found that netropsin was not selectively cytocidal to only HMGA2-overexpressing colorectal cancer cells [193], and its potential binding to other AT-hook DNA domains might introduce off-target side-effects, making it therapeutically inefficient. Rather, an antihelminthic drug niclosamide has been repurposed and is selective against HMGA2-overexpressing colorectal cancer cells, reversing the *HMGA2*-driven gene signature, and inhibiting cell cycle-related genes in these cells [193]. Analogously, successful treatment of poorly differentiated thyroid carcinoma with tyrosine kinase inhibitor selumetinib and histone deacetylase inhibitor panobinostat led to the significant downregulation of HMGA2 expression, correlating with an upregulation or stable expression of associated miRNA *let-7b* [194].

## Figures and Tables

**Figure 1 ijms-21-03151-f001:**
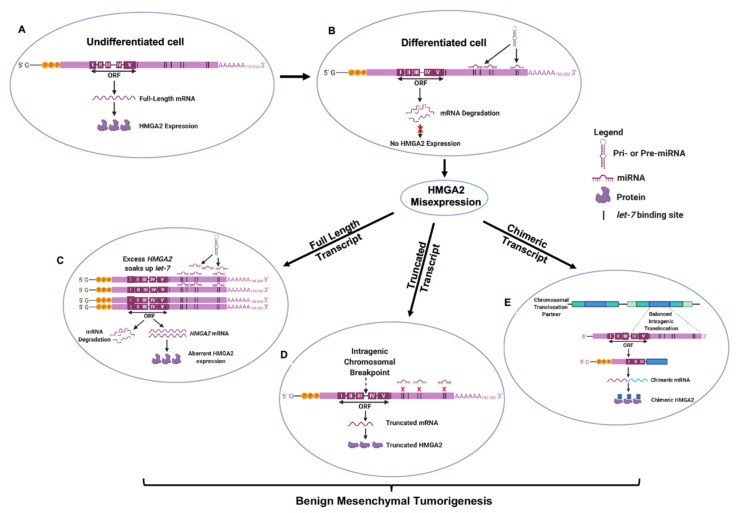
*let-7* regulatory mechanisms govern HMGA2 misexpression, driving benign mesenchymal tumorigenesis. (**A**) Absence of tumor suppressor let-7 miRNA expression in the undifferentiated mesenchyme ensures ubiquitous HMGA2 expression during mammalian embryogenesis. (**B**) Increased *let-7* expression as mesenchymal tissues mature and differentiate inhibits HMGA2 expression by binding to sites in the 3’ UTR of HMGA2 mRNA leading to transcript degradation. (**C**) Abundant *HMGA2* transcription in many uterine leiomyomas and mammary fibroadenomas yield enough mRNA possessing multiple 3’ UTR binding sites that can soak up available *let-7* allowing for remnant HMGA2 misexpression of full-length transcripts. (**D**) Intragenic chromosomal breaks at preferred locations between exons III and IV and (**E**) balanced translocations with chromosomal partners forming chimeric *HMGA2* fused to ectopic sequences, are two mechanisms ensuring loss of *HMGA2* 3’ UTR and its let-7 binding sites allowing for misexpression of full-length *HMGA2* transcripts and activation of the HMGA2 pathway in differentiated benign mesenchymal tumors.

**Table 1 ijms-21-03151-t001:** Balanced Chromosomal Rearrangements Forming Hmga2-Induced Gene Fusions in Human Mesenchymal Tumors.

Tumor Type	[Refs]	Chromosomal Translocation Partners	Chromosomal Rearrangement	Fusion Protein
Lipoma	[29,30,43]	1	t(1;12)(p32;q14)	HMGA2/PPAP2B
	[77,78]	2	t(2;12)(q37;q14)	HMGA2/ACKR3
	[29,39,69,70,71,72,73,74]	3	t(3;12)(q28;q14)	HMGA2/LPP
	[29,43,73,79]	5	t(5;12)(q33;q15)	HMGA2/EBFEBF/BC058822
	[79]	5	t(5;12)(q33;q14)	HMGA2/EBF1
	[29,79]	5	ins(5;12)(q33;q14q21)	Partial Genomic Loss
	[29,73,79]	5	t(5;12)(q33;q15)	No Genomic Loss/Gain
	[79]	5	t(5;12)(q32;q14)	No Genomic Loss/Gain
	[79]	5	t(5;12)(q32;q15)	No Genomic Loss/Gain
	[29,80]	5	ins(12;5)(q15;q33q13)	No Genomic Loss/Gain
	[81,82,83]	9	t(9;12)(p22;q14)	HMGA2/NFIB
	[42]	Intragenic	t(12;12)(q14;q14)	HMGA2/GRIP1
	[38]	13	t(12;13)(q14;q13)	HMGA2/LHFP
	[51]	15	t(12;15)(q14;q24)	HMGA2/Ser-Thr domain
	[42]	18	t(12;18)(q14;q12)	HMGA2/SETBP1
	[42]	18	t(12;18)(q14~q15;q12~q21)	HMGA2/SETBP1
	[42]	18	t(12;18)(q14~q15;q12~q21)	HMGA2/GRIP1
	[42]	18	t(12;18)(q14~q15;q12~q21)	HMGA2/18q12.3 Sequence
Osteochondrolipoma	[42]	18	t(12;18)(q14~q15;q12~q21	HMGA2/SETBP1
Uterine Leiomyoma	[50]	7	t(7;12)(q31;q14)	HMGA2/COG5
	[39,55,56]	8	t(8;12)(q22;q14)	HMGA2/COX6C
	[84]	Intragenic	der(12)(q14)	HMGA2/RTVL-H
	[85,86,87]	14	t(12;14)(q14;q24)	RAD51l1/HMGA2
	[54]	14	t(12;14)(q15;q11)	HMGA2/HE110
Soft Tissue Chondroma	[37]	3	t(3;12)(q27;q15)	HMGA2/LPP
Sarcoma	[76]	Intragenic	t(12;12)(q15;q14)	FRS2/HMGA2
	[76]	1	t(1;12)(p32;q14)	HMGA2/DAB1
	[76]	Intragenic	t(12;12)(q14;q13)	HMGA2/PCBP2
	[76]	Intragenic	t(12;12)(q14;q12)	HMGA2/NELL2
	[76]	Intragenic	t(12;12)(q14;q21)	HMGA2/PPFIA2
	[76]	1	t(1;12)(p32;q14)	HMGA2/C1orf87
	[76]	Intragenic	t(12;12)(q14;q13)	HMGA2/SARNP
	[76]	11	t(11;12)(p11;q14)	HMGA2/ARFGAP2
	[76]	Intragenic	t(12;12)(q14;q22)	HMGA2/NR2C1
	[76]	6	t(6;12)(q24;q14)	UTRN/HMGA2
Myolipoma	[88]	9	t(9;12)(p22;q14)	HMGA2/C90RF92
Aggressive angiomyxoma	[89]	1	t(1;12)(p32;q15)	HMGA2(3’UTR)/NT032977.8
Extra-skeletal Osteochondroma	[90]	Intragenic	inv(12)(p12q14)	HMGA2/SOX5
Spindle Cell Sarcoma	[91]	Intragenic	t(12;12)(q14;q15)	HMGA2/DYRK2
Pulmonary Chondroid Harmatoma (PCH)	[68]	3	t(3;12)(q27;q14;q15)	HMGA2/LPP

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
