# Peer review of "High Mobility Group AT-Hook 2 (HMGA2) Oncogenicity in Mesenchymal and Epithelial Neoplasia"

_ijms, 2020, doi:10.3390/ijms21093151_

Round 1

Reviewer 1 Report

Comments on Manuscript ID: ijms-777176: “High Mobility Group AT-Hook 2 (HMGA2) Oncogenicity in Mesenchymal and Epithelial Neoplasia" by Unachukwu et al.

The authors studied the role of HMGA2 misexpression contribution to mesenchymal tumorigenesis and benign vs metastatic distinction and role of miRNA in HMGA2 misexpression in tumors-very nice review compiling the overall updated studies about HMGA2 misexpression.

I have following recommendations before the publication of this MS, which need to be addressed in MS:

  1. Line 43-51: HMGA2 is expressed during embryonic development, but declining expression in developmental stages later and distributed to connective tissues of mesenchymal origin, as observed in mice, means HMGA2 has a role in development. However, the author’s claim of misexpression of HGM2A in differentiated tissues of mesenchymal origin compared to canonical expression in undifferentiated tissues (line 94-95) is in contrast to the studies showing overexpression of HMGA2 led to EMT, activation of TGF beta pathway as well as epithelial tumor metastasis (line 74-76). Does that mean “the terminology of misexpression” is actually mosaic expression having genetic alterations leading to unstable mRNA transcripts as well as full transcripts skipping let7 miRNA suppression? That in-turn would led me to Fig.1C which needs clear relabeling of miRNA binding transcripts leading to mRNA degradation and non-binding transcripts leading to full transcripts, thus mosaic expression as well as subheading 2.1 which need re-writing in context to misexpression.
  2. Line 32-34 as well as subheading 2.2, author described that HMGA2 was found to be nuclear as it binds to AT rich sequences of DNA. However, as reported HMGA1 has a cytosolic counterpart-which was found to be involved in its secretion having invasive capability and also has a mitochondrial counterpart. The whole MS is lacking in providing the clue whether there is any such counterpart for HMGA2 and if yes, what is the role of that counterpart in context to misxpression especially in mesenchymal tissue origins or lymphangioleiomyomatosis (LAM) if any?
  3. The role of let7 miRNA in decreasing the expression of HGMA2 which have different effects on various tumor types. The author should include any study which knocks out/down this miRNA and see if the HMGA2 shows misexpression due to chromosomal translocation alterations and if that leads to EMT transitions? That information should be included in the MS.

The paper is technically sound and well written. comments above- will make the MS better.

Author Response

Reviewer 1 Comments:

The authors studied the role of HMGA2 misexpression contribution to mesenchymal tumorigenesis and benign vs metastatic distinction and role of miRNA in HMGA2 misexpression in tumors-very nice review compiling the overall updated studies about HMGA2 misexpression.

I have following recommendations before the publication of this MS, which need to be addressed in MS:

  1. Line 43-51: HMGA2 is expressed during embryonic development, but declining expression in developmental stages later and distributed to connective tissues of mesenchymal origin, as observed in mice, means HMGA2 has a role in development. However, the author’s claim of misexpression of HGM2A in differentiated tissues of mesenchymal origin compared to canonical expression in undifferentiated tissues (line 94-95) is in contrast to the studies showing overexpression of HMGA2 led to EMT, activation of TGF beta pathway as well as epithelial tumor metastasis (line 74-76). Does that mean “the terminology of misexpression” is actually mosaic expression having genetic alterations leading to unstable mRNA transcripts as well as full transcripts skipping let7 miRNA suppression? That in-turn would led me to Fig.1C which needs clear relabeling of miRNA binding transcripts leading to mRNA degradation and non-binding transcripts leading to full transcripts, thus mosaic expression as well as subheading 2.1 which need re-writing in context to misexpression.

Response:

We realize that terminologies used to describe canonical and abnormal expression of HMGA2 in both mesenchymal and epithelial type-specific tumorigenic tissues could be confusing. Notwithstanding the ubiquity of its use in literature, we have replaced the term “overexpression” with “ectopic expression” in  the instances where it pertains to HMGA2 expression in epithelial tumors (Lines 66, 161, 248, 398, 434). “Overexpression” was meant to denote instances when a gene (HMGA2) is expressed in a cell type of a lineage where it is not normally expressed (differentiated epithelial tissues), and we have now changed it ectopic expression to convey our thoughts more clearly. Misexpression on the other hand, represents expression of the gene (HMGA2) in a cell type where it is not normally expressed (differentiated mesenchymal tissue), within the same lineage (mesenchymal lineage)

  1. Line 32-34 as well as subheading 2.2, author described that HMGA2 was found to be nuclear as it binds to AT rich sequences of DNA. However, as reported HMGA1 has a cytosolic counterpart-which was found to be involved in its secretion having invasive capability and also has a mitochondrial counterpart. The whole MS is lacking in providing the clue whether there is any such counterpart for HMGA2 and if yes, what is the role of that counterpart in context to misxpression especially in mesenchymal tissue origins or lymphangioleiomyomatosis (LAM) if any?

Response:

After literature review, there seems to be a dearth in knowledge regarding intracellular HMGA2 translocation in mesenchymal tumorigenesis and studies are warranted in this regard. We added the following excerpt to lines 453-471 in section 3 to summarize our findings. “Dynamic intracellular localization processes of HMGA2 mRNA transcripts and translation products have been reported to also modulate epithelial tumorigenesis and metastatic potential. HMGA2 expression has been detected in the cell membrane of quiescent non-transformed, post-natal mouse keratinocytes but upon onset of proliferation, membrane-to-nuclear translocation of the protein was observed [1]. Analogous HMGA2 nuclear translocation was also been observed upon ex vivo culture of mouse keratinocytes and during cutaneous carcinogenesis in DMBA and TPA mouse models where HMGA2 induces its own expression in an autoregulatory loop by binding to the Hmga2 promoter [1]. Normal epithelial prostate cell also exhibited low levels of HMGA2 expression in the plasma membrane which switched to predominantly cytoplasmic then nuclear localization with increasing prostate tumor grade, metastatic potential, and HMGA2 expression [2]. Specific small non-coding circular RNAs (circNSUN2) promotes EMT and human and mouse colorectal cancer (CRC) cell metastasis by stabilizing high levels of HMGA2 cytoplasmic RNA-protein interactions in a circNSUN2/IGF2BP2/HMGA2/ complex and activating the HMGA2 pathway. In this study, lower levels of cytoplasmic HMGA2 mRNA was associated with lower liver metastasis in CRC [3]. HMGA2 mRNA cytoplasmic stability is similarly governed by higher order ribonucleoprotein complexes it forms with RNA binding proteins such as those of the IMP family necessary strongly correlated in solid cancers. Moreover, HMGA2 mRNA is physically associated with IMP3 RNP in vivo [4]. There is need for studies of these HMGA2 intracellular translocation processes and their relevance in mesenchymal tumorigenesis.

  1. The role of let7 miRNA in decreasing the expression of HGMA2 which have different effects on various tumor types. The author should include any study which knocks out/down this miRNA and see if the HMGA2 shows misexpression due to chromosomal translocation alterations and if that leads to EMT transitions? That information should be included in the MS.

Response: Upon literature review we could not find any such studies correlating let-7 miRNA knockdown to HMGA2 misexpression induced chromosomal translocations. If we had hitherto this review, we would have certainly included it in our manuscript write-up to support our postulates. We agree such investigations are pertinent to establishing the role of let-7 as a regulon of HMGA2 misexpression. We did report a study (Reference 168) where let-7a knockdown in nasopharyngeal carcinoma cells suppressed EMT and tumor cell proliferation in Lines 363-364, section 3.

To place higher emphasis to this reference study type, we have changed Lines 363-364 from “while let-7A performs an analogous inhibitory function in nasopharyngeal carcinomas induced by HMGA2 expression” to “while let-7a knockdown exhibited an analogous inhibitory function in nasopharyngeal carcinomas induced by HMGA2 expression, decreasing HMGA2 and the expression of EMT marker genes Snail, Slug, and vimentin.”

The paper is technically sound and well written. comments above- will make the MS better.

 References

  1. Li, Y.; Pi, X.-y.; Boland, K.; Lad, S.; Johnson, K.; Verfaillie, C.; Morris, R. J., Hmga2 translocation induced in skin tumorigenesis. Oncotarget 2017, 8, (18), 30019-30029.
  2. Hawsawi, O.; Henderson, V.; Burton, L. J.; Dougan, J.; Nagappan, P.; Odero-Marah, V., High mobility group A2 (HMGA2) promotes EMT via MAPK pathway in prostate cancer. Biochemical and biophysical research communications 2018, 504, (1), 196-202.
  3. Chen, R.-X.; Chen, X.; Xia, L.-P.; Zhang, J.-X.; Pan, Z.-Z.; Ma, X.-D.; Han, K.; Chen, J.-W.; Judde, J.-G.; Deas, O.; Wang, F.; Ma, N.-F.; Guan, X.; Yun, J.-P.; Wang, F.-W.; Xu, R.-H.; Dan, X., N6-methyladenosine modification of circNSUN2 facilitates cytoplasmic export and stabilizes HMGA2 to promote colorectal liver metastasis. Nature Communications 2019, 10, (1), 4695.
  4. Degrauwe, N.; Suvà, M.-L.; Janiszewska, M.; Riggi, N.; Stamenkovic, I., IMPs: an RNA-binding protein family that provides a link between stem cell maintenance in normal development and cancer. Genes & Development 2016, 30, (22), 2459-2474.

Reviewer 2 Report

This article aims to review the mechanisms of HMGA2-induced tumorigenesis in mesenchymal and epithelial tumors but it fails to be exhaustive and often lacks of clarity. As it is structured, the article should focus on the different molecular mechanisms downstream of HMGA2 in both mesenchymal and epithelial tumors. However, it provided an excellent analysis of the cytogenetic rearrangements of the HMGA2 gene in mesenchymal tumors, then focusing on the consequent upstream let-7-mediated regulation of HMGA2 misexpression, whereas only little space is dedicated to how HMGA2 induces transformation of mesenchymal cells. Very interesting and novel is the focus on LAM. It should be emphasized in the title that should change to: High Mobility Group AT-Hook2 (HMGA2) Oncogenicity in Mesenchymal and Epithelial Neoplasia: focus on Lymphangioleiomyomatosis. Overall the manuscript needs a substantial revision. Suggestions for improvement are given below:

Major issues

  1. Abstract should mention the focus on LAM
  2. Section 2 appears twisted because it introduces let-7 from the beginning and only later on well explain its role with a dedicated subsection. Furthermore, the titles of the subsections are partially redundant and misleading. For a better clarity this section should be better rearranged as follows:
    • Figure 1 should go after the first paragraph of the subsection 2.1.4
    • The title of subsection 2.1 should be “HMGA2 misexpression”
    • Lines 167-181 (from “High levels” to the end); 194-197 (from “Truncations” to “ (Figure 1D)”) should be deleted. These concepts can be explained in the following subsection 2.1.4
    • The title of subsection 2.2 should be “Effects of HMGA2 misexpression on Chromatin structure”
    • Section 4 should be placed here as subsection 2.3 entitled: “HMGA2-induced tumorigenesis in Lymphangioleiomyomatosis”
  3. Subsection 2.1.1 is quite misleading. The cited paper (56) is an in vitro study in murine fibroblasts and it cannot be compared with in vivo studies in humans (8, 40, 41, 57, 58). Furthermore, studies with transgenic mice (17) are cited together with cytogenetic studies in humans without explaining important details. For a better clarity, please rephrase the text as follows:
  • Line 157-159 “Despite previous studies had shown that truncation and/or addition of ectopic sequences after the third AT-hook were necessary for neoplastic transformation of murine fibroblasts (56), many human uterine leiomyomata, mammary fibroadenomas, and salivary gland adenomas were found to harbor full-length HMGA2 misexpression (8, 40, 41, 57, 58).
  • Line 167: After ref. 41, add: “Consistently, misexpression of full-length Hmga2 induces benign mesenchymal tumors in mice (17)”
  1. lines 260-262: The concept that by soaking up all available let-7, HMGA2 subtracts it to its suppressive activity on many other oncogenes is a very important mechanism explaining the tumorigenicity of the full length HMGA2. Please, expand it.
  2. Another important study that should be discussed in this section is the work cited as reference 26, in which the HMGA2 misexpression has been studied in mesenchymal stem cells, i.e. the proposed cells of origin of lipomas and uterine leyomiomas
  3. Section 3 should be expanded as well. I understand that the authors believe EMT is one of the major mechanisms responsible of HMGA2 tumorigenesis in epithelial tumors, and I agree, but other mechanisms, including roles in cell cycle, apoptosis, cancer stem cells and others have been explored and should be mentioned in this section.
  4. Lines 376-381 express important concepts about differences between HMGA2-induced mesenchymal and epithelial tumorigenicity. They should be further developed and go as a separate section after sections 2 and 3.

Minor issues

  1. Line 28: the correct chromosome localization of the human HMGA2 gene is 12q14.3 (Wei JJ. HMGA2 (high mobility group AT-hook 2) Atlas Genet Cytogenet Oncol Haematol 2016; 20(7): 403-412)
  2. Line 36: please correct as follows “exhibit” and “yield”
  3. Lines 88 and 95: cite a reference for the expression of HMGA2 in undifferentiated mesenchyme and differentiated tissue; line 94: delete reference to Figure 1A
  4. Supplementary table 1: it is not clear what the last column includes. According to its labeling title the resulting fusion protein is indicated in some cases. In other cases “none” is reported. Do you mean that no fusion protein is formed? So, what about the empty table cells? What is the difference with “none”? Finally, in some other cases there is indication of genomic loss/gain. Please, clarify the content of this column and label it appropriately.
  5. Line 101: 12q15 should be 12q13-15
  6. Lines 105 and 338: please check well the meaning of “severally” if appropriate
  7. Lines 105-109: Please check the font size
  8. Figure 1: some font size is too small
  9. Lines188-189: please rephrase as follows “Human HMGA23’ UTR is 2.9kb long and harbors up to….”
  10. Table 1: Why some chromosomal rearrangements are in bold? Why there are some question marks? If they are not typo errors, please indicate the meaning in a legend
  11. Line 269: rephrase as follows: “overexpression of HMGA2, which is associated to increasing…”
  12. Line 341: please correct as follows “in vitro”
  13. Lines 343-345: “ This activation of the TGFb pathway…..” is quite misleading because seems to suggest that TGFb is downstream of HMGA2, whereas it is the opposite. Please rephrase for a better clarity
  14. Lines 419-420: change “reported in Section B (I)” with “reported above”
  15. Line 427: change “Hmga2-let-7” with “let-7-Hmga2 axis”
  16. Lines 475-477: if no aknowledgments needed, please delete all this section

Author Response

Reviewer 2 Comments

This article aims to review the mechanisms of HMGA2-induced tumorigenesis in mesenchymal and epithelial tumors but it fails to be exhaustive and often lacks of clarity. As it is structured, the article should focus on the different molecular mechanisms downstream of HMGA2 in both mesenchymal and epithelial tumors. However, it provided an excellent analysis of the cytogenetic rearrangements of the HMGA2 gene in mesenchymal tumors, then focusing on the consequent upstream let-7-mediated regulation of HMGA2 misexpression, whereas only little space is dedicated to how HMGA2 induces transformation of mesenchymal cells. Very interesting and novel is the focus on LAM. It should be emphasized in the title that should change to: High Mobility Group AT-Hook2 (HMGA2) Oncogenicity in Mesenchymal and Epithelial Neoplasia: focus on Lymphangioleiomyomatosis. Overall the manuscript needs a substantial revision. Suggestions for improvement are given below:

Major issues

  1. Abstract should mention the focus on LAM

Response: Abstract has been revised to include our focus on HMGA2-mediated neoplasms in LAM.

2. Section 2 appears twisted because it introduces let-7 from the beginning and only later on well explain its role with a dedicated subsection. Furthermore, the titles of the subsections are partially redundant and misleading. For a better clarity this section should be better rearranged as follows:

  • Figure 1 should go after the first paragraph of the subsection 2.1.4

Response: Figure 1 has been moved to a location after line 219 in subsection 2.1.4

  • The title of subsection 2.1 should be “HMGA2 misexpression”

Response: The title of subsection 2.1 has been changed from “HMGA2 Misexpression in Mesenchymal Tumors” to “HMGA2 Misexpression”

  • Lines 167-181 (from “High levels” to the end); 194-197 (from “Truncations” to “ (Figure 1D)”) should be deleted. These concepts can be explained in the following subsection 2.1.4

Response: Lines 167-175 and 194-197 have been deleted from subsection 2.1.2.

The following aspect of Lines 167-175 was added to Lines 234-236 in subsection 2.1.4:

“High levels of HMGA2 transcription in larger leiomyomata could also encode more 3’ UTR binding sites to soak up available mature let-7 miRNA limiting the repressive and degradative function of let-7 on HMGA2 expression (40, 41), allowing for greater rates of tumorigenesis compared to smaller sized uterine fibroids.”

  • The title of subsection 2.2 should be “Effects of HMGA2 misexpression on Chromatin structure”

Response: The title of subsection 2.2 has been changed from “Alteration of Chromatin Structure” to “Effects of HMGA2 Misexpression on Chromatin Structure”

  • Section 4 should be placed here as subsection 2.3 entitled: “HMGA2-induced tumorigenesis in Lymphangioleiomyomatosis”

Response: Section 4 has been moved up to subsection 2.3 and the subtitle renamed as “HMGA2-induced tumorigenesis in Lymphangioleiuomyomatosis”. Given this change, “Section 5” has now been renumbered to “Section 4.”

3. Subsection 2.1.1 is quite misleading. The cited paper (56) is an in vitro study in murine fibroblasts and it cannot be compared with in vivo studies in humans (8, 40, 41, 57, 58). Furthermore, studies with transgenic mice (17) are cited together with cytogenetic studies in humans without explaining important details. For a better clarity, please rephrase the text as follows:

  • Line 157-159 “Despite previous studies had shown that truncation and/or addition of ectopic sequences after the third AT-hook were necessary for neoplastic transformation of murine fibroblasts (56), many human uterine leiomyomata, mammary fibroadenomas, and salivary gland adenomas were found to harbor full-length HMGA2 misexpression (8, 40, 41, 57, 58).

Response: Lines 157-159 has been changed from “Contrary to previous postulates that truncation of HMGA2 and/or addition of ectopic sequences after the third AT-hook were necessary for mesenchymal tumorigenicity, many uterine leiomyomata, mammary fibroadenomas, and salivary gland adenomas were found to harbor full-length Hmga2 coding regions in mice and humans” to “Despite previous studies showing that truncation and/or addition of ectopic sequences after the third AT-hook were necessary for neoplastic transformation of murine fibroblasts, many human uterine leiomyomata, mammary fibroadenomas, and salivary gland adenomas were found to harbor full-length HMGA2 mRNA transcripts.” References citing cytogenetic studies of HMGA2 expression in mice has been separated from human studies.

  • Line 167: After ref. 41, add: “Consistently, misexpression of full-length Hmga2 induces benign mesenchymal tumors in mice (17)”
  • Response: In Line 167 after ref. 41, “Consistently, misexpression of full-length Hmga2 induces benign mesenchymal tumors in mice (17)” has been added.
  • 4. lines 260-262: The concept that by soaking up all available let-7, HMGA2 subtracts it to its suppressive activity on many other oncogenes is a very important mechanism explaining the tumorigenicity of the full length HMGA2. Please, expand it.

Response: In lines 241-244, added “High levels of HMGA2 transcription in larger leiomyomata could also encode more 3’ UTR binding sites to soak up available mature let-7 miRNA limiting the repressive and degradative function of let-7 on HMGA2 expression (40, 41), and allowing for greater rates of tumorigenesis compared to smaller sized uterine fibroids.”

In lines 246-252, added “This mechanism could also account for slight differences in tumor spectrum mediated by full length forms of HMGA2 versus truncated HMGA2 forms lacking the let-7 binding sites in mice (18, 96). Ectopic expression of truncated HMGA2 transgenes in an immortalized mesenchymal stem-like cell line stymied adiopogenic differentiation and upregulated genes for transcription and intracellular protein transfer compared to wildtype HMGA2 indicative of higher tumorigenicity and lesser let-7 repressive activity in cells expressing truncated HMGA2 (97).”   

5. Another important study that should be discussed in this section is the work cited as reference 26, in which the HMGA2 misexpression has been studied in mesenchymal stem cells, i.e. the proposed cells of origin of lipomas and uterine leyomiomas

Response: In lines 276-284 in section 2.1.4, after “…determining HMGA2 expression/misexpression.”, added “This seemingly limited influence of let-7 in directly regulating HMGA2 misexpression could be more reflective of its cooperative roles in establishing benign mesenchymal tumorigenic phenotypes. In mechanistic studies using adipogenic progenitor cells of lipomas and uterine leiomyomas, a p14Arf-MDM2-TP53-let-7 network stabilized HMGA2 expression, lowering the tendency for malignant transformation while maintaining stem-like proliferative characteristic of benign mesenchymal tumor cells (104). p14Arf represses HMGA2 via a TP53 mechanism that has been linked to increased expression of let-7 (105), whereas FGF1-stimulated increase in HMGA2 increased p14Arf indicative a bimodal mechanism for regulating cellular senescence and stem-like cell renewal as the molecular basis for benign mesenchymal tumorigenesis (9, 104).   

6. Section 3 should be expanded as well. I understand that the authors believe EMT is one of the major mechanisms responsible of HMGA2 tumorigenesis in epithelial tumors, and I agree, but other mechanisms, including roles in cell cycle, apoptosis, cancer stem cells and others have been explored and should be mentioned in this section.

Response: In lines 434-452, this paragraph was added, “Ectopic expression of HMGA2 can also promote epithelial tumor cell proliferation and metastasis by influencing the cell cycle in a tumor cell-type dependent manner, where for instance, its knockdown arrests ovarian cancer cells at G1 (1) and G2/M arrest for leukemia cells (2). In some instances, HMGA2 can exert tumorigenic effects on cell cycle by directly inducing cyclin A2 (3), activating protein-1 (AP1) expression (4), and MDM2-mediated p53 ubiquitination (5) facilitating cell proliferation. Indirectly, HMGA2 can activate phosphatidylinositide 3-kinase (PI3K)/AKT/mTOR/p70S6k signaling which inhibits tumor suppressors p16INK4A, p21CIP1/WAF1 (6), retinoblastoma protein (pRB) (7) and p14Arf (8) to enable transitions through cell cycle checkpoints, facilitate cell proliferation and restrain cellular senescence. During these replicative cycles, HMGA2 has also been postulated to either inhibit DNA damage response (DDR) mechanisms that ensure genomic stability at replication forks, leading to increased DNA mutational rates at onset of tumorigenesis, or augment these DDR mechanisms to reduce replication recovery times after replication fork arrest in stem and cancer cells (9, 10). HMGA2 can also mediate epithelial tumorigenesis by modulating apoptosis in cancer cells. Breast adenocarcinoma cells overexpressing HMGA2 exhibit less apoptotic events compared to cells with low HMGA2 expression by a mechanism involving HMGA2-mediated inhibition of miR-34a and subsequent de-repression of Bcl-2 promoting an anti-apoptotic pathway (11). HMGA2 also protects cancer cells from apoptosis by hyperactivating the PI3K/Akt pathway which impairs activation of caspase-9 and Bad in a gastric cancer cell line (12).

7. Lines 376-381 express important concepts about differences between HMGA2-induced mesenchymal and epithelial tumorigenicity. They should be further developed and go as a separate section after sections 2 and 3.

Response: We purposefully summarized the differences between HMGA2-induced mesenchymal and epithelial tumorigenicity as stated because to state anything more definitive, comprehensive comparative studies between both tumorigenic mechanisms will need to be performed. We resist being too speculative given the dearth of data determining these differences.

Minor issues

8. Line 28: the correct chromosome localization of the human HMGA2 gene is 12q14.3 (Wei JJ. HMGA2 (high mobility group AT-hook 2) Atlas Genet Cytogenet Oncol Haematol 2016; 20(7): 403-412)

Response: Human HMGA2 gene locus has been changed from “chromosome 12 at band q15” to “chromosome 12 at band q14.3”. Associated citation has been updated to “Wei JJ. HMGA2 (high mobility group AT-hook 2) Atlas Genet Cytogenet Oncol Haematol 2016; 20(7): 403-412”

9. Line 36: please correct as follows “exhibit” and “yield”

Response: In line 36, “exhibits” has been changed to “exhibit”. “Yields” has been changed to “yield.”

10. Lines 88 and 95: cite a reference for the expression of HMGA2 in undifferentiated mesenchyme and differentiated tissue; line 94: delete reference to Figure 1A

Response: In line 88, references inserted describing HMGA2 expression in both mouse and human development and in differentiated tissues:

  • Hirning-Folz U, Wilda M, Rippe V, Bullerdiek J, Hameister H. The expression pattern of the Hmgic gene during development. Genes, Chromosomes and Cancer 1998; 23: 350-357.
  • Gattas GJ, Quade BJ, Nowak RA, Morton CC. HMGIC expression in human adult and fetal tissues and in uterine leiomyomata. Genes, chromosomes & cancer 1999; 25: 316-322.

In line 94, reference inserted describing HMGA2 expression in human development and in differentiated tissues:

  • Gattas GJ, Quade BJ, Nowak RA, Morton CC. HMGIC expression in human adult and fetal tissues and in uterine leiomyomata. Genes, chromosomes & cancer 1999; 25: 316-322.

Reference to Figure 1A has been deleted

11. Supplementary table 1: it is not clear what the last column includes. According to its labeling title the resulting fusion protein is indicated in some cases. In other cases “none” is reported. Do you mean that no fusion protein is formed? So, what about the empty table cells? What is the difference with “none”? Finally, in some other cases there is indication of genomic loss/gain. Please, clarify the content of this column and label it appropriately.

Response: We intended to highlight chromosomal rearrangements leading to chimeric HMGA2 mRNS transcripts and as Table 1 in the manuscript also provides such information, the last column titled “Fusion Protein” in Supplementary Table 1 has been deleted for clarity.

Line 101: 12q15 should be 12q13-15

Response: In line 101, changed “12q15” to “12q13-15”

Lines 105 and 338: please check well the meaning of “severally” if appropriate

Response: In line 105, “severally” was deleted as the same message could be conveyed without the word. In line 338, changed “EMT has severally been reported…” to “In many studies, EMT has been reported…” to convey the same message with more clarity.

Lines 105-109: Please check the font size

Response: In lines 105-109, reduced font sizes from 11 to 10

Figure 1: some font size is too small

Response: All font size labeling images in Figure 1 have been drastically increased in size.

Lines188-189: please rephrase as follows “Human HMGA23’ UTR is 2.9kb long and harbors up to….”

Response: In lines 188-189, changed Human HMGA2 3’ UTR of is 2.9kb long harboring up to…” to “Human HMGA23’ UTR is 2.9kb long and harbors up to….”

Table 1: Why some chromosomal rearrangements are in bold? Why there are some question marks? If they are not typo errors, please indicate the meaning in a legend

Response: All boldened fonts in Table 1 have been adjusted to regular fonts. All detailed systems for designating structural chromosomal aberrations used in Table 1 have been changed to the short system for consistency where the chromosomal rearrangement found in extra-skeletal osteochondroma was changed from “inv(12)(qter->q14~15::p11->q13::q14~15->q13::p11->pter) [13]/idem,t(5;13)(q13;p11)” to “inv(12)(p12q14)” and that for spindle cell sarcoma was changed from “del(X)(p?11p?22), der(12)(12pter→12q?22::12q?15→q?22::16p11→ 16pter),-16,+r(12).” to “t(12:12)(q14;q15)”. This is for consistency of pathogenic genotypes listed in the table, and in accordance with the ISCN rules for listing chromosomal rearrangements (Curr Protoc Hum Genet 2001; Appendix 4: Appendix 4C.) and as published in the Mitelman Database for Chromosomal Aberrations and Gene Fusions in Cancer (Mitelman Database of Chromosome Aberrations and Gene Fusions in Cancer (2020). Mitelman F, Johansson B and Mertens F (Eds.), https://mitelmandatabase.isb-cgc.org) 

Line 269: rephrase as follows: “overexpression of HMGA2, which is associated to increasing…”

Response: In line 269, changed “overexpression of HMGA2 associated increasing” to “expression of HMGA2, which is associated with increasing…”

Line 341: please correct as follows “in vitro”

Response: In line 341, “invitro” has been changed to “in vitro”

Lines 343-345: “ This activation of the TGFb pathway…..” is quite misleading because seems to suggest that TGFb is downstream of HMGA2, whereas it is the opposite. Please rephrase for a better clarity

Response: For clarity in lines 343-345, “ This activation of the TGFb pathway was found to be necessary for HMGA2-induced epithelial tumorigenesis and invasiveness via EMT” has been changed to “For HMGA2 to induce epithelial tumorigenesis and invasiveness via EMT, it is necessary that TGFb pathway is activated”

Lines 419-420: change “reported in Section B (I)” with “reported above”

Response: In lines 419-420, “reported in Section B (I)” has been changed to “reported above”

Line 427: change “Hmga2-let-7” with “let-7-Hmga2 axis”

Response: In line 427, “Hmga2-let-7” has been changed to “let-7-Hmga2 axis”

Lines 475-477: if no aknowledgments needed, please delete all this section

Response: Lines 475-477 have been deleted

Round 2

Reviewer 2 Report

The manuscript has been improved. I haven't any more comments.